

# Bone lesions in Chinese POEMS syndrome patients: imaging characteristics and clinical implications

Fengdan Wang[1,*], Xufei Huang[2,3,*], Yan Zhang[1], Jian Li[3], Daobin Zhou[3] and Zhengyu Jin[1]

[1] Department of Radiology, Peking Union Medical College Hospital, Beijing, China
[2] Peking Union Medical College, Beijing, China
[3] Department of Hematology, Peking Union Medical College Hospital, Beijing, China
[*] These authors contributed equally to this work.

Corresponding authors
Jian Li, LiJian@pumch.cn
Zhengyu Jin, pumchradiology@126.com

## ABSTRACT

**Objective.** Bone lesion is crucial for diagnosing and management of polyneuropathy, organomegaly, endocrinopathy, monoclonal protein, and skin change (POEMS) syndrome, a rare plasma cell disorder. This study is to compare the effectiveness of X-ray skeletal survey (SS) and computed tomography (CT) for detecting bone lesions in Chinese POEMS syndrome patients, and to investigate the relationship between bone lesion features and serum markers.

**Methods.** SS and chest/abdomen/pelvic CT images of 38 Chinese patients (26 males, 12 females, aged 21–70 years) with POEMS syndrome recruited at our medical center between January 2013 and January 2015 were retrospectively analyzed. Bone lesions identified by CT were further categorized according to the size (<5 mm, 5–10 mm, >10 mm) and appearance (osteosclerotic, lytic, mixed). The percentage of plasma cells in bone marrow smears, type of immunoglobulin, platelet (Plt), and levels of serum bone metabolic markers and inflammatory factors including alkaline phosphatase (ALP), calcium, phosphate, parathyroid hormone (PTH), beta-isomerized C-telopeptide ($\beta$-CTx), vascular endothelial growth factor (VEGF), and interleukin (IL)-6 levels were also recorded.

**Results.** Of the 38 POEMS syndrome patients, the immunoglobulin heavy chain isotypes were IgA in 25 patients (65.8%; 25/38) and IgG in 13 patients (34.2%; 13/38), and the light chain isotypes were λ in 35 patients (92.1%; 35/38) and κ in 3 patients (7.9%; 3/38). There were 23 patients with thrombocytosis. More patients with bone lesions were detected by CT than by SS (97.4% vs. 86.8%). The most commonly affected location was the pelvis (89.5%), followed by the spine, clavicle/scapula/sternum/ribs, skull, and long bones. Of the 38 POEMS syndrome patients, 35 (94.6%) had osteosclerotic and 32 (86.5%) had mixed lesions. Osteosclerotic lesions were typically scattered, variable in size, and plaque-like, whereas mixed lesions were pouch-shaped or soup bubble-like with a clear sclerotic margin and were generally larger. Although the majority of bone lesions were small in size, 23 (62.2%) had at least one lesion >10 mm. There was no correlation between serum marker levels and bone lesion patterns after Bonferroni correction (all $P > 0.001$).

**Conclusions.** CT is more sensitive and accurate than SS in detecting bone lesions in POEMS syndrome.

## INTRODUCTION

Polyneuropathy, organomegaly, endocrinopathy, monoclonal protein, and skin change (POEMS) syndrome is a rare plasma cell disorder. Early diagnosis is challenging since the syndrome is rare and can easily be mistaken for other disorders. Both mandatory major criteria (polyneuropathy and monoclonal plasma cell proliferative disorder), one of the three major criteria (Castleman disease, osterosclerotic lesions, elevated serum levels of plasma vascular endothelial growth factor (VEGF)), and at least one of the six minor criteria are required for a definitive diagnosis. The diagnostic criteria of the Mayo Clinic were revised in 2007, when the presence of pathognomonic osteosclerotic lesion was upgraded from minor to major criterion (*Li & Zhou, 2013*).

Bone lesions are important not only for diagnosing and evaluating POEMS syndrome, but are also crucial for administering appropriate treatments (*Dispenzieri, 2014*). Focal radiation is recommended for isolated bone lesions, but systemic chemotherapy should be used in cases of multiple lesions. Therefore, imaging is indispensable for accurately diagnosing POEMS syndrome and determining the treatment approach that is required. However, there is no consensus on the best imaging modality for identifying bone lesions (*Minarik et al., 2012*). X-ray skeletal survey (SS) and computed tomography (CT) are the two most commonly used methods in clinical practice, but there have been few systematic investigations comparing the effectiveness of the two methods. Most studies are case reports or have examined a small patient sample (*Glazebrook et al., 2015*; *Shi et al., 2015*; *Shibuya et al., 2011*), in part due to the rarity and complexity of this disorder.

Although bone lesions involve diffuse infiltration of light chain-restricted plasma cells (*Nakajima et al., 2007*), POEMS patients present with neither bone pain nor pathologic fractures, which is a major feature of POEMS syndrome that distinguishes it from multiple myeloma. However, the clinical implications of bone lesions and whether they correspond to disease severity or the levels biochemical markers of bone metabolism or inflammatory factors remain open questions.

In this study, we compared the effectiveness of SS and CT for investigating bone lesions in Chinese POEMS syndrome patients. The relationship between bone lesion features from CT images and serum marker levels was also examined.

## METHODS

### Patient population

This was a retrospective study approved by Institutional Review Board, thus the requirement for informed consent was waived. All Chinese patients diagnosed with POEMS syndrome in our medical center from January 2013 to January 2015 were assessed for eligibility. Inclusion criteria were as follows: (a) patients met the 2007 diagnostic criteria for POEMS syndrome; (b) patients underwent an X-ray SS for bone lesions; (c) a chest/abdomen/pelvic CT scan was performed within 1 month of SS; and (d) all SS and CT examinations were completed before treatment.

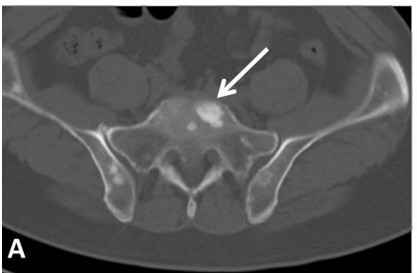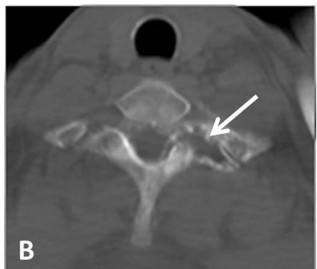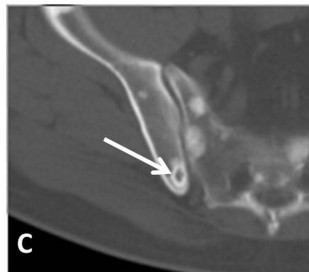

**Figure 1** **Three types of bone lesions illustrated in CT images.** (A) Osteosclerotic lesion; (B) Osteolytic lesion; (C) Mixed lesion.

The percentage of plasma cells in bone marrow smears as well as type of immunoglobulin, platelet (Plt), alkaline phosphatase (ALP), serum calcium, phosphate, parathyroid hormone (PTH), beta-isomerized C-telopeptide ($\beta$-CTx), VEGF, and interleukin (IL)-6 levels were also recorded. The patients' personal information was de-identified prior to analysis.

### Image scanning and interpretation

The conventional SS included skull, cervical/thoracic/lumbar spine, pelvis, bilateral humeri and femora. The CT scan range was from the upper margin of sternum to the ischium, and the slice thickness was 7 mm. Two radiologists (FD.W. with six years' experience in general diagnostic radiology and Y.Z. with 15 years' experience in musculoskeletal specialization) evaluated the bone lesions on a consensus basis using the Centricity PACS system (GE Healthcare, Barrington, IL, USA). When CT images were evaluated, the bone window was adjusted to a width of 2000 HU and a level of 350 HU. The number of bone lesions was recorded by location.

Lesions revealed in CT images were categorized according to size (<5 mm, 5–10 mm and >10 mm) and appearance (lytic, sclerotic, and mixed). Size was measured based on maximum lesion diameter. According to previous studies (*Glazebrook et al., 2015*), an osteosclerotic lesion had high density surrounded by normal bone marrow; an osteolytic lesion had low density; and a mixed lesion had a central lytic component and was surrounded by a sclerotic margin (Fig. 1).

### Statistical analysis

The Paired $t$ test was used to compare the number of bone lesions revealed by SS and CT. A $P$ value <0.05 was considered statistically significant. The Pearson correlation was used to evaluate the relationship between the number of bone lesions and serum marker levels. Bonferroni correction was achieved when $P$-values were less than 0.001 (0.05/42) in the correlation analysis. Data were analyzed using SPSS v.18.0 software (SPSS Inc., Chicago, IL, USA).

## RESULTS

### Study population

From January 2013 to January 2015, a total of 38 Chinese patients diagnosed with POEMS syndrome according to the 2007 diagnostic criteria were included: 26 males and 12 females, with a mean age of 50.5 years (range: 21–70 years). The immunoglobulin heavy chain

isotypes were IgA in 25 patients (65.8%; 25/38) and IgG in 13 patients (34.2%; 13/38), and the light chain isotypes were λ in 35 patients (92.1%; 35/38) and κ in 3 patients (7.9%; 3/38). All the 38 patients underwent bone marrow biopsy, and their Plt, serum calcium, phosphate, ALP, PTH, $\beta$-CTx, and VEGF levels were examined. Serum IL-6 levels were evaluated in 25 patients. The average percentage of plasma cells in bone marrow biopsies was 1.0% (range: 0%–5.5%). Levels of other serum markers were as follows: Plt $335 \times 10^9$/L (range: $89$–$628 \times 10^9$/L); ALP 82.7 U/L (range: 50–135 U/L); calcium, 2.13 mmol/l (range: 1.80–2.40 mmol/l); phosphate, 1.42 mmol/l (range: 1.10–1.90 mmol/l); PTH, 30.16 g/ml (range: 3.0–101.0 g/ml); $\beta$-CTx, 1.16 ng/ml (range: 0.30–3.50 ng/ml); VEGF, 5,972 ng/l (range: 534–14,328 ng/l); and IL-6, 6.4 pg/ml (range: 2.0–16.0 pg/ml). There were 23 patients with thrombocytosis, while none of the ALP was elevated.

## SS vs. CT

Of the 38 Chinese patients diagnosed with POEMS syndrome, 33 (86.8%) were found to have 276 bone lesions by SS. The most commonly affected location was the pelvis (71.1%; 27/38), with up to 135 lesions identified in this bone; this was followed by spine (47.3%, 18/38), cranium (36.5%, 14/38), and extremities (28.9%, 11/38). An analysis of chest/abdomen/pelvis CT images revealed 37 patients (97.4%) with lesions in these bones. The scope of chest/abdomen/pelvis CT scans included only thoracic/lumbar spine, pelvis, and other flat bones (clavicle/scapula/sternum/ribs). Within this scope, there was no patient whose bone lesion was detected by X-ray but not by CT. The total number of bone lesions detected by CT was 994, which was far greater than the number detected by SS (276 lesions). As detected by SS, the pelvis was the most commonly affected location in CT scans (89.5%; 34/38). The number of patients with lesions in the thoracic or lumbar spine or other bones was 20 (52.6%), 19 (50.0%), and 20 (52.6%), respectively, based on CT scans. The distribution of bone lesions by both methods is shown in Fig. 2.

Since the thoracic and lumbar spine and pelvis were covered by both SS and CT, the number of patients with bone lesions detected by each of these imaging methods was compared. At all three locations, more bone lesions were detected by CT than by SS (thoracic spine, $P = 0.001$; lumbar spine, $P = 0.003$; pelvis, $P = 0.001$) (Fig. 3). In some cases, even after bone lesions were identified in CT images it was not possible to detect the lesions in SS films obtained from the same patient (Fig. 4). CT was especially useful for detecting small bone lesions and clearly showing their outlines.

## Characteristics of bone lesions in CT images

The size of each bone lesion in CT images was measured in order to further characterize bone manifestations of POEMS syndrome. Nearly 90% of patients had bone lesions smaller than 10 mm (<5 mm: 86.5%, 32/38; 5–10 mm: 89.2%, 33/38) (Fig. 5A). However, 23 patients (62.2%) had lesions >10 mm; the largest lesion was 76 mm (Fig. 6). Over half (62.8%, 624/994) of lesions were <5 mm and 14.3% (142/994) were >10 mm (Fig. 5B).

Of the 38 POEMS syndrome patients, 35 (94.6%) had osteosclerotic and 32 (86.5%) had mixed lesions (Fig. 7A). Osteosclerotic lesions were typically scattered, variable in size, and plaque-like (Fig. 8), whereas mixed lesions were pouch-shaped or soup bubble-like with a

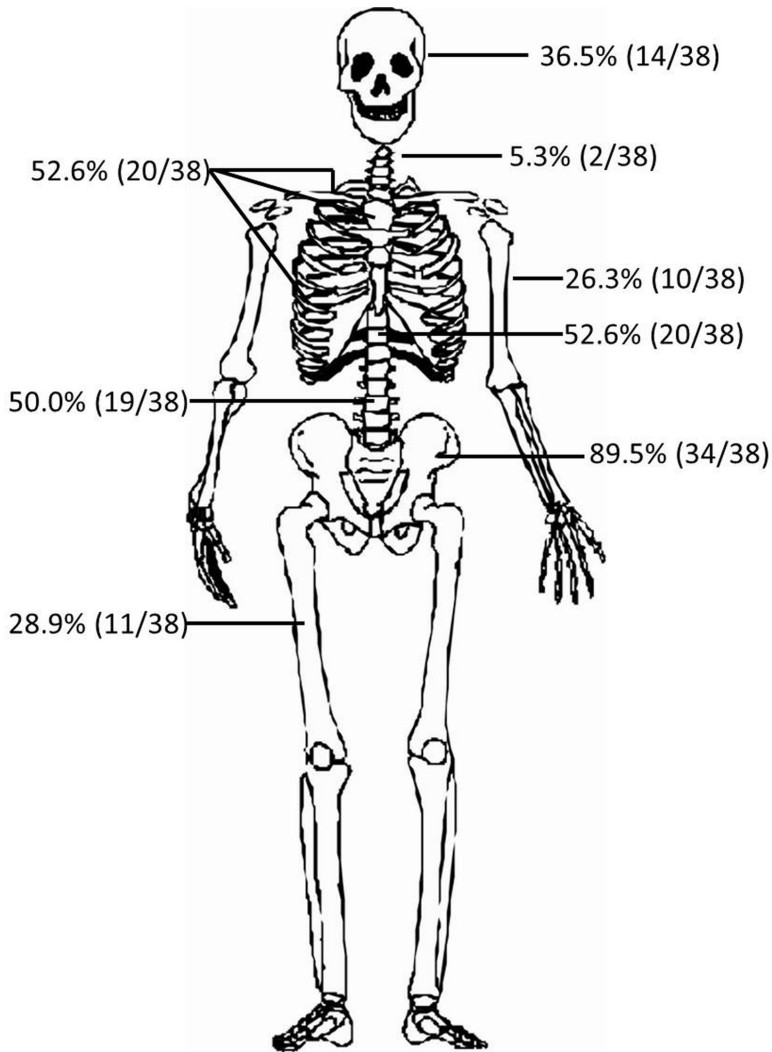

**Figure 2  Distribution of bone lesions in POEMS syndrome.** The pelvis was the most commonly affected region, followed by spine, clavicle/scapula/sternum/ribs, cranium, and long bones (humeri and femora). Percentages of patients with affected cranium, cervical spine, and long bones were based on the SS; the others were based on CT images.

clear sclerotic margin and were generally larger (Fig. 9). Patients with pure osteolytic lesions accounted for 15.8% (6/38) of cases. Osteosclerotic lesions were the most prevalent type of bone lesions, accounting for 76.9% (764/994) of the total, followed by mixed (22.5%) and osteolytic (0.6%) types (Fig. 7B).

## Relationship between bone lesions and serum marker levels

The relationship between bone lesions and the percentage of plasma cells in bone marrow, as well as serum marker levels (calcium, phosphate, PTH, $\beta$-CTx, VEGF, and IL-6) were detected. However, after Bonferroni correction, none of these markers had a significant correlation with the number, size, nor type of the  bone lesions. (Table 1)

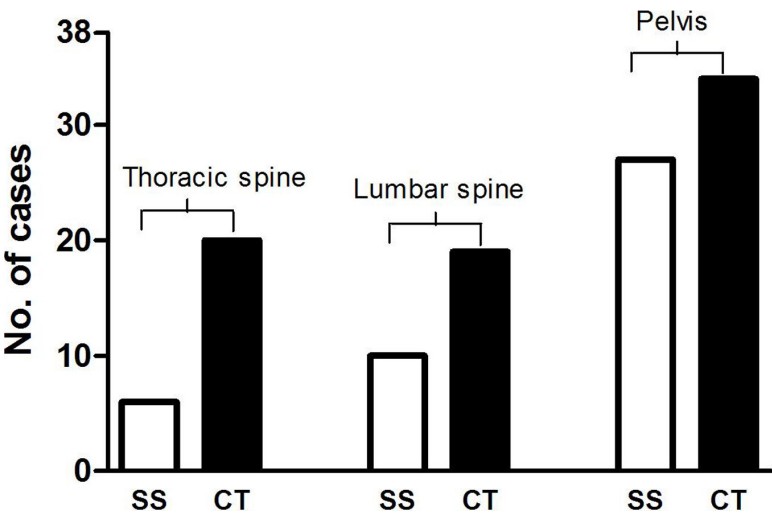

**Figure 3 Comparison of the number of patients with bone lesions revealed by SS and CT.**

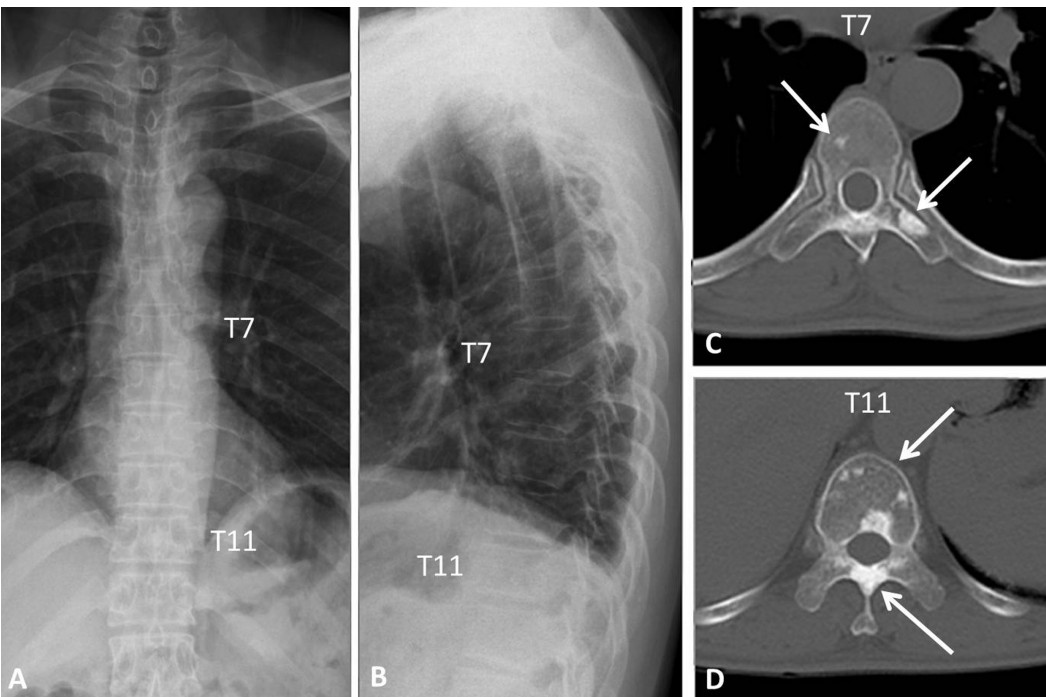

**Figure 4 Thoracic spine of a 54-year-old male with POEMS syndrome.** Antero-posterior (A) and lateral thoracic (B) spine X-rays were interpreted as normal. Sclerotic lesions of the vertebral body and appendix (C, D) were clearly observed by CT (arrows).

## DISCUSSION

This study retrospectively analyzed bone lesions of 38 Chinese POEMS syndrome patients who underwent SS and chest/abdomen/pelvis CT scans. CT identified more patients and detected more bone lesions than SS (97.4% vs. 86.8% and 994 vs. 276

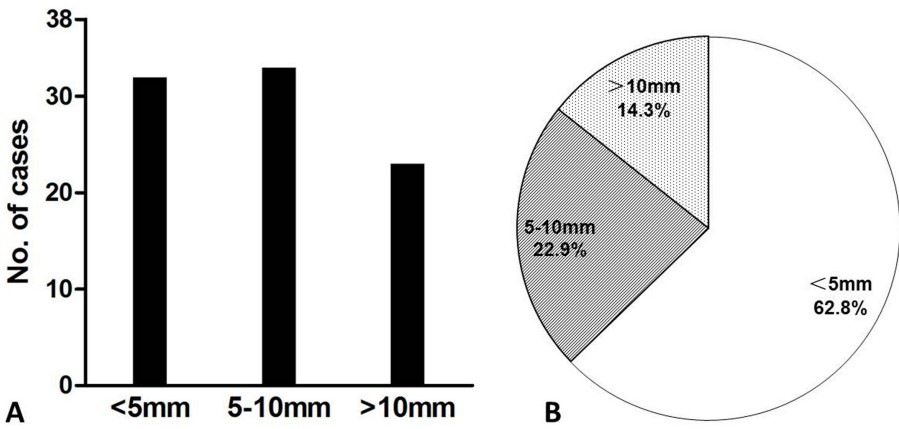

**Figure 5** Number of patients (A) and bone lesions (B) with respect to lesion size.

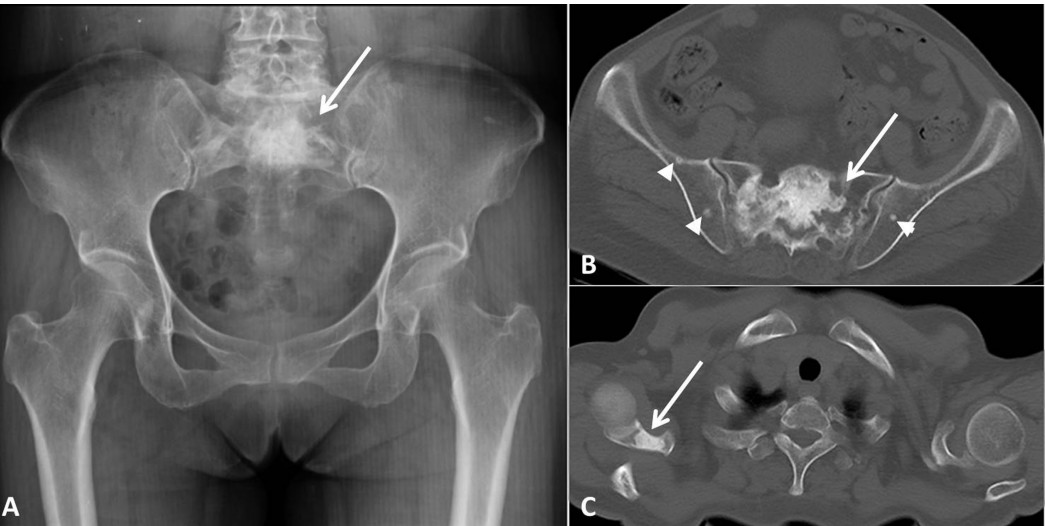

**Figure 6** Massive bone lesion of a 56-year-old female with POEMS syndrome. A large (76-mm diameter) mixed lesion was detected in the sacrum (A and B, arrows). The osteolytic component of the lesion was clearly visible in the CT image (B). Several small osteosclerotic lesions in the ilium were also more evident in the CT image (B, arrow heads). A sclerotic lesion was detected in the right scapula (C, arrow).

lesions, respectively). Moreover, smaller lesions were detected by CT, which revealed details of the lesions and delineated their margins. Although most (90%) bone lesions were <10 mm, 62.2% of patients had at least one bone lesion >10 mm. Similar numbers of patients had osteosclerotic and mixed lesions (94.6% and 86.5%, respectively). There was no correlation between bone lesion features determined from images and serum levels of metabolic markers or inflammatory factors.

Our finding that, within the thoracic/abdominal/pelvic region, CT is superior to SS for detecting bone lesions is consistent with previous studies. The results suggest that CT may play a dominant role in diagnosis of POEMS, and may replace SS for evaluation of the disease in the future. One study of 24 POEMS patients examined by both CT and SS found a false negative rate of 36% using the latter method (*Glazebrook et al., 2015*), while another study

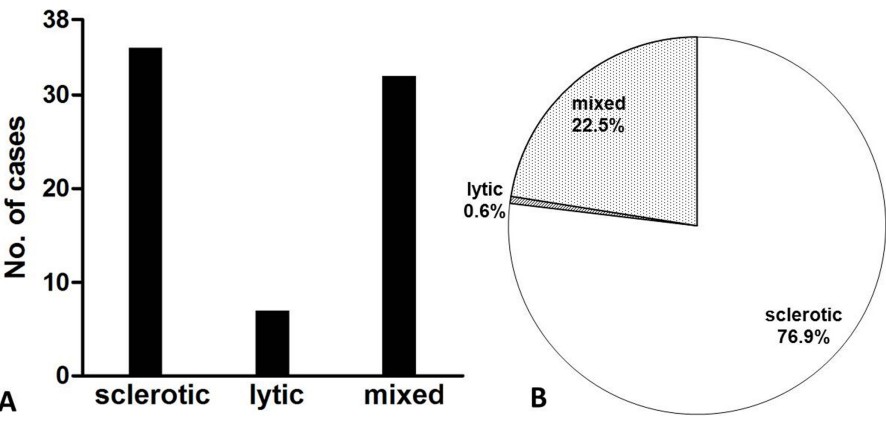

**Figure 7** Number of patients (A) and bone lesions (B) as a function of lesion appearance.

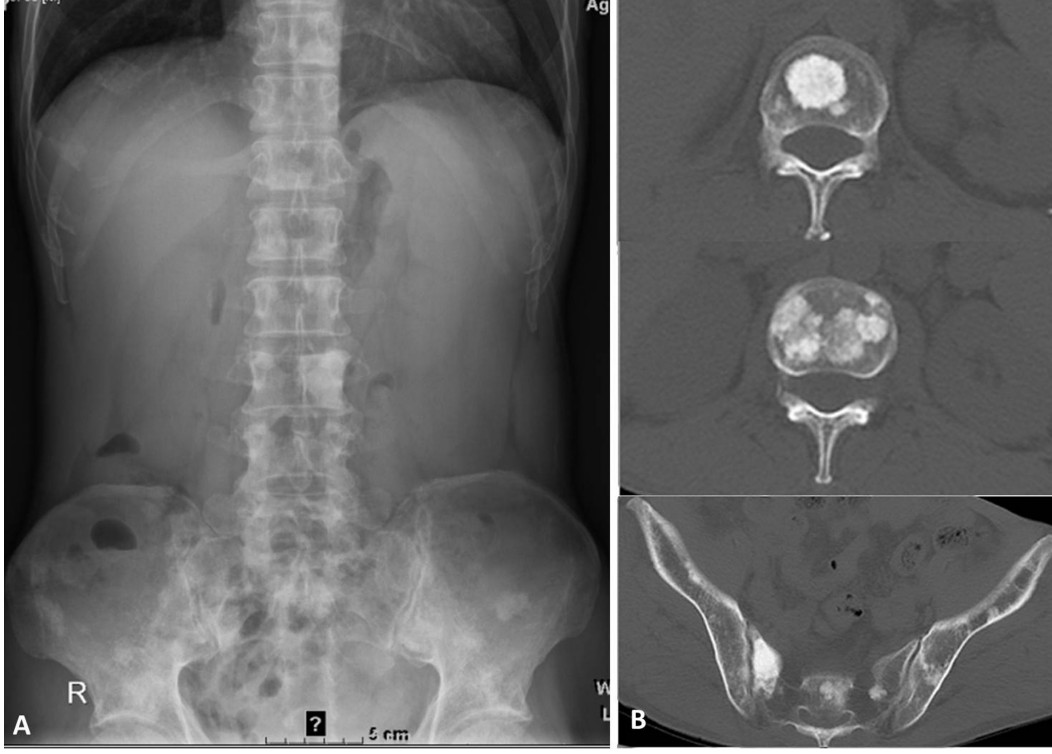

**Figure 8** Osteosclerotic lesions in the spine and pelvis of a 53-year-old male with POEMS syndrome. X-ray (A) and CT (B) images showing multiple plaque-like sclerotic bone lesions in the lumbar spine, sacrum, and ilium.

arrived at similar conclusions by confirming that CT was able to detect small bone lesions that were not apparent in the SS (*Shi et al., 2015*). Some Chinese researchers have reported a much lower rate of affected bones (27%–55%) (*Cao et al., 2014*; *Cui et al., 2011*; *Li et al., 2011*; *Zhang et al., 2010*) than studies from Western countries (86%–96%) (*Dispenzieri & Buadi, 2013*; *Dispenzieri et al., 2003*). This discrepancy may partially be attributable to the fact that only X-ray scans were used in the Chinese studies.

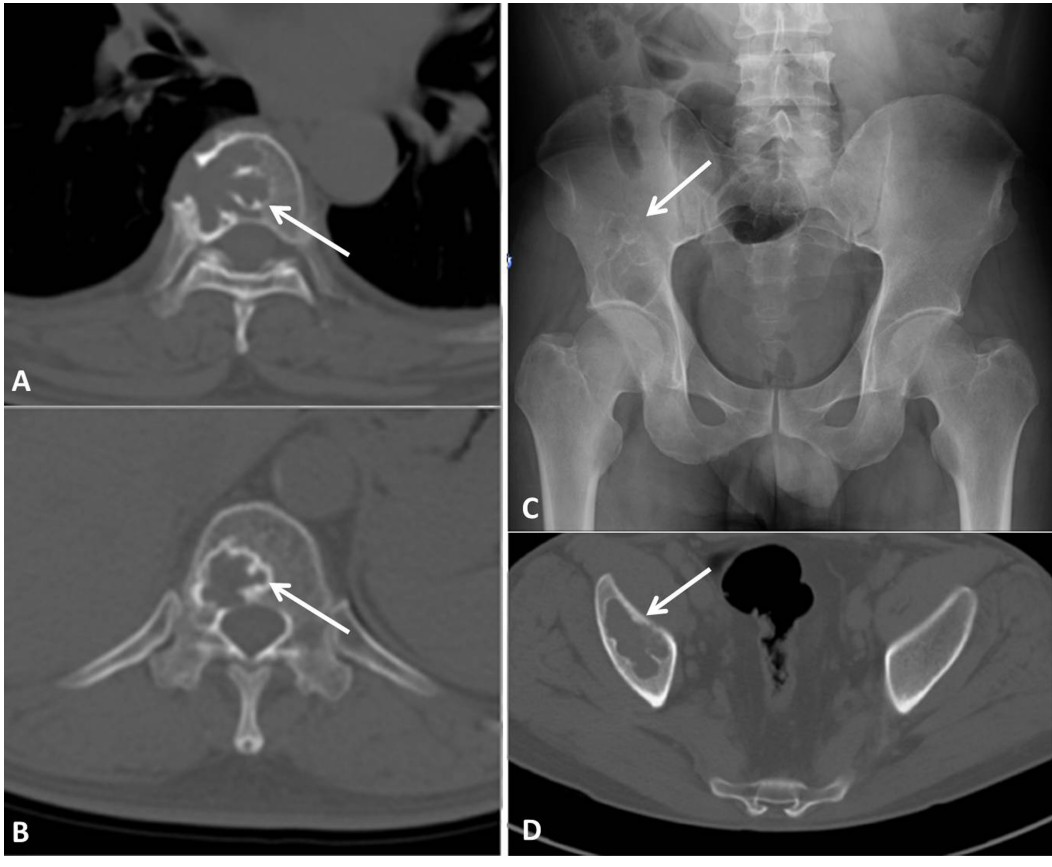

**Figure 9** **Typical mixed bone lesions of POEMS syndrome.** Mixed lesions in the thoracic vertebrae (A, B) and right ilium (C, D) showing typical pouch and soup-bubble like appearance with a clear sclerotic rim (arrows). Note the broken bone cortex of the affected vertebral body (A).

**Table 1** **Correlation between lab markers and bone lesion features in POEMS syndrome.**

| Lab markers | Mean ± SD | P value with bone lesion features [b] | | | | | |
|---|---|---|---|---|---|---|---|
| | | Total | <5 mm | 5–10 mm | >10 mm | Sclerotic lesion | Mixed lesion |
| Plasma cells[a] (%) | 0.96 ± 1.23 | 0.785 | 0.987 | 0.509 | 0.568 | 0.873 | 0.692 |
| Calcium (mmol/l) | 2.13 ± 0.15 | 0.824 | 0.876 | 0.620 | 0.997 | 0.802 | 0.148 |
| Phosphate (mmol/l) | 1.42 ± 0.18 | 0.723 | 0.705 | 0.864 | 0.673 | 0.676 | 0.932 |
| PTH (g/ml) | 30.16 ± 20.3 | 0.752 | 0.592 | 0.906 | 0.974 | 0.959 | 0.393 |
| $\beta$-CTx (ng/ml) | 1.16 ± 0.66 | 0.518 | 0.600 | 0.408 | 0.521 | 0.507 | 0.866 |
| VEGF (ng/ml) | 5,972 ± 3,371 | 0.284 | 0.209 | 0.410 | 0.676 | 0.153 | 0.581 |
| IL-6 (pg/ml) | 6.4 ± 4.0 | 0.401 | 0.419 | 0.292 | 0.711 | 0.316 | 0.889 |

**Notes.**

[a]The percentage of plasma cells in bone marrow biopsies.

[b]P value between lab markers and bone lesion features. After Bonferroni correction, P-values of less than 0.001(0.05/42) were considered statistically significant. The relationship between lab markers and osteolytic lesions was not analyzed here since there were only six patients with this kind of bone lesion.

PTH, parathyroid hormone; $\beta$-CTx, beta-isomerized C-telopeptide; VEGF, vascular endothelial growth factor; IL-6, interleukin 6.

This finding may have some clinical implications. First, CT may identify bone lesions which SS is not able to show, which constitute a major criterion of POEMS syndrome. Therefore, this will help to diagnose the rare disorder especially for some complicated cases. Second, if SS detects isolated bone lesion, focal radiation is the choice. However, if CT identifies more bone lesions than SS, the treatment strategy should be changed to systemic chemotherapy (*Humeniuk et al., 2013*). Nevertheless, it is unfeasible to conduct whole body CT scans because of concerns of radiation dose; yet it is also unnecessary to obtain a SS if chest/abdomen/pelvis CT is used concurrently. POEMS syndrome patients must undergo a chest/abdomen/pelvis CT in order to determine whether other features of the disorder such as extravascular overload, organomegaly, and enlarged lymph nodes are also present (*Shi et al., 2016*). We therefore suggest that a CT bone window is always used for POEMS patients to identify bone lesions, while X-ray SS is used adjunctively to detect lesions specifically in the skull and long bones. Furthermore, as CT could detect more and smaller bone lesions than X-ray, it seems not reasonable to give radiation to a patient with only one or two small (<10 mm) bone lesions. Therefore, the treatment strategy used nowadays which is mainly based on the number of bone lesions might need to be reconsidered.

Though the majority of bone lesions are osteosclerotic and <10 mm, mixed and larger lesions require more attention. An early study described prominent osseous proliferation in the spine of three POEMS syndrome patients (*Resnick et al., 1981*), but this is seldom seen nowadays since patients are diagnosed and treated much earlier than before as a result of advances in diagnostic techniques. Osteosclerotic lesions are pathognomonic for POEMS syndrome and are one of the major diagnostic criteria (*Dispenzieri, 2014*). However, we agree with other reports that mixed lesions with a soap-bubble or pouched appearance are also typical of this disorder (*Glazebrook et al., 2015*). Therefore, mixed lesions could also be included as diagnostic criteria in the future.

In this study we found that up to 86.5% of the patients had at least one mixed lesion, which means that Chinese patients share the similar bone lesion features of Western ones. Two previous Chinese studies did not report any mixed lesions with a soap-bubble or pouched appearance (*Shi et al., 2015*; *Shi et al., 2016*), possibly because of the small number of patients (22 and 24 patients), some of whom were included in both studies. Mixed lesions also tend to be larger than other types, with the central lytic component showing increased intensity on T2-weighted sequences of magnetic resonance imaging (MRI) (*Nakayama-Ichiyama et al., 2012*), and higher activity in 18F-fluorodeoxyglucose positron emission tomography (PET) (*Glazebrook et al., 2015*; *Minarik et al., 2012*). This may aid in the selection of an appropriate treatment and monitoring of disease progression.

The relationship between bone lesions and clinical markers of POEMS syndrome is not well understood. Osteosclerotic bone lesions are characterized by the diffuse infiltration of light chain-restricted plasma cells (*Nakajima et al., 2007*; *Nakayama-Ichiyama et al., 2012*). Bone marrow aspirates and biopsies from 87 patients with POEMS syndrome demonstrated a constellation of $\gamma$-restricted monoclonal gammopathy plasma cell rims around lymphoid aggregates in bone marrow (*Dao et al., 2011*); however, most bone marrow samples were from bones without lesions. It has been hypothesized that VEGF, a marker of POEMS syndrome, is expressed by osteoblasts and causes new bone formation (*Dispenzieri & Buadi,*

_2013_). One study found that 61.0% of patients with high levels of serum VEGF (>2,000 ng/l) had osteosclerotic lesions (_Cao et al., 2014_), but this study employed only SS and may therefore have underestimated the number of lesions. Only three patients in our study had serum VEGF levels <2,000 ng/l; one patient exhibited no bone lesions but had a VEGF level of 4,965 ng/l. These results suggest that VEGF level is unrelated to the type and number of bone lesions.

This study had several limitations. Firstly, the number of patients was still relatively small due to the rarity of POEMS syndrome; however, our patient sample had more homogeneity than in most previous studies. Secondly, the pathology of bone lesions was not examined. Since different types of bone lesion have distinct pathological features, this requires further investigation. Thirdly, only thoracic-abdominal-pelvic bone lesions were included in the correlation analysis, which may not represent the total bone disease burden. Lastly, other imaging methods such as MRI, skeletal scintigraphy and PET-CT were not included in our analysis, but their effectiveness relative to X-ray SS and CT should also be considered.

## CONCLUSION

CT was more sensitive and accurate in detecting bone lesions in POEMS syndrome patients than X-ray SS, but the latter method was useful for identifying lesions in the skull and extremities. Finally, bone lesions in POEMS syndrome were not associated with the levels of metabolic markers or inflammatory factors.

### Funding

The Beijing Natural Science Foundation (grant No. 7142130), the Specialized Research Fund for the Doctoral Program of Higher Education (grant No. 2013110611000), the Capital Health Research and Development of Special (grant No. 2011-4001-03), Peking Union Medical College New Star (2011, for LJ) and the National Natural Science Foundation of China (grant No. 81570195) supported this research. The funders had no role in study design, data collection and analysis, decision to publish, or preparation of the manuscript.

### Grant Disclosures

The following grant information was disclosed by the authors:
The Beijing Natural Science Foundation: 7142130.
Specialized Research Fund for the Doctoral Program of Higher Education: 2013110611000.
Capital Health Research and Development of Special: 2011-4001-03.
Peking Union Medical College New Star: 2011, for LJ.
National Natural Science Foundation of China: 81570195.

### Competing Interests

The authors declare there are no competing interests.

## Author Contributions

- Fengdan Wang conceived and designed the experiments, performed the experiments, analyzed the data, wrote the paper, prepared figures and/or tables, reviewed drafts of the paper.
- Xufei Huang conceived and designed the experiments, performed the experiments, analyzed the data, prepared figures and/or tables, reviewed drafts of the paper.
- Yan Zhang performed the experiments, prepared figures and/or tables, reviewed drafts of the paper.
- Jian Li conceived and designed the experiments, analyzed the data, contributed reagents/materials/analysis tools, wrote the paper, prepared figures and/or tables, reviewed drafts of the paper.
- Daobin Zhou contributed reagents/materials/analysis tools, reviewed drafts of the paper.
- Zhengyu Jin conceived and designed the experiments, contributed reagents/materials/-analysis tools, reviewed drafts of the paper.

## Human Ethics

The following information was supplied relating to ethical approvals (i.e., approving body and any reference numbers):

This was a retrospective study approved by Institutional Review Board of Peking Union Medical College Hospital, thus the requirement for informed consent was waived.

## Data Availability

The raw data has been supplied as Supplemental Dataset.

## Supplemental Information

Supplemental information for this article can be found online at http://dx.doi.org/10.7717/peerj.2294#supplemental-information.

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
