# Peer review of "Bone lesions in Chinese POEMS syndrome patients: imaging characteristics and clinical implications"

_PeerJ, doi:10.7717/peerj.2294_

## Round 0.1 · original submission · Major Revisions

· Academic Editor

Major Revisions

The reviewers and I all conclude that this is an interesting scientific study that is well written. The reviewers have several relevant remarks that require attention.

My remark is as follows: This manuscript describes a retrospective study comparing SS versus CT scanning of POEMS patients. It seems like you have been using an additional modality for research purposes. Is there any IRB approval or equivalent for this study?

·

Basic reporting

The authors have revealed CT is more useful to detect bone lesions in POEMS syndrome than X-ray. Small bone lesions are more common, compared to larger. This manuscript is well-written.
I have some questions about this article. Authors should answer these.

Experimental design

#1. The authors should clearly write the range of X-ray imaging. Did X-ray imaging include lower thigh and foot? Does “long bones” mean thighbone?

#2. Do the authors have data of platelet and type of immunoglobulin (Ig-G or Ig-A)? The authors should analyse the relationship between these and bone lesion features.

#3. Did the authors find patients with bone lesion which was detected by X-ray but not by CT? To compare the usefulness of these imaging methods, the author should disclose this.

Validity of the findings

#4. The authors have written that mixed lesion is typical in POEMS syndrome, but they also have written mixed lesion was 22.5%. I recommend the authors delete these sentences.

#5. Patients had bone lesions in 36.5% skull and 28.9% thigh. The authors should also mention the usefulness of X-rays for skull and thigh.

Additional comments

No comments.

·

Basic reporting

No comments.

Experimental design

No Comments.

Validity of the findings

No comments.

Additional comments

The article “Bone lesions in Chinese POEMS syndrome patients: imaging characteristics and clinical implications” prepared by Wang et al. is interesting and well-founded. However, it seems that the main viewpoint and results, “CT was more sensitive and accurate in detecting bone lesions in POEMS syndrome patients than X-ray SS, but the latter method was useful for identifying lesions in the skull and extremities. The most prevalent lesion type was osteosclerotic.”, are not very novel since there are some similar publications previously (See the discussion). Another finding “bone lesions in POEMS syndrome were not associated with the levels of metabolic markers or inflammatory factors” is a negative one which has small possibility to be utilized in clinic work. However, the patient sample is large and the data are relatively detailed and persuasive.
Additionally, the difference of the SS and CT partially may be from the sites, besides the sensitivity. X-ray is better than CT in diagnosis of bone lesions in long bone and skull, where the SS can cover large area but are out of field of CT. SS is not as same sensitive as CT in diagnosis of bone lesions in spine, where overlapping of figures makes it difficult for SS to detect the lesions. These considerations should be included in the discussion.
The paper is written very well.

·

Basic reporting

This article is well written, clear and to the point. The research seems to have been conducted soundly. Their results clearly show the superiority of CT compared to SS in detecting lesions in POEMS syndrome

Experimental design

Comparing the blood parameters to bone lesions. Why was the Alkaline Phosphatase not added to the list? This is a much more sensitive biomarker for bone turnover in sclerotic lesions (like in prostate cancer). It can be suspected that it will correlate better with the number and size of lesions.

The authors used a CT slice thickness of 7 mm. This is very thick. They however divide the lesions < 5 mm, 5-10 mm and > 10 mm. With such thick slices, it can therefore be expected that lesions < 5 mm and some lesions 5-10 mm will be missed.

Validity of the findings

Is this study clinically useful?
Firstly, it is mentioned that for clinical usefulness, the imaging modality should help in making the diagnosis. The authors state that in 33 out of 38 patients SS made the diagnosis while this was 37/38 for CT. So clearly 4 patients were only diagnosed with CT. The authors however mention in line 202 that “POEMS syndrome patients must undergo a chest/abdomen/pelvis CT in order to determine
whether other features of the disorder such as extravascular overload, organomegaly enlarged lymph nodes are also present”. Since patients then routinely undergo CT the question should be raised if at all additional SS is warranted?

The authors however failed to mention important findings that may help in determining if at all SS is useful. They mention that 89% of patients had pelvic bone lesions with decreasing frequency in lumbar, thoracic spine and ribs etc. The least prevalent sites are skull and long bones. The question will now be? If only a pelvis and spine CT was performed, are any patients missed due to lesions that were ONLY prevalent in skull/longbones? If this is the case, the authors should rather attempt to publish these findings than the current article.

A second clinical application of their findings: POEMS patients with a single skeletal lesion should undergo radiation, those with multiple lesions, systemic therapy. Again, it would be of great interest to know what the clinical usefulness of finding 994 with CT compared to 276 lesions? If > 2 lesions are found, systemic therapy is indicated, so finding 20, 30 or 100 will not change the management at all? Again the SS vs CT relevance can be questioned. How many of the patients in this study only had one lesion? I presume none. It would be of interest to know (apart from the 4 patients that the skeletal survey missed) did the fewer lesions found in SS compared to CT result in any change in management?

Additional comments

So in summary, the authors did show that CT is vastly better that SS. However the clinical usefulness of this is questionable. As mentioned above, does finding more lesions on CT change the management at all? The authors are encourage rather to focus on determining what minimum imaging protocol will suffice to a) aid in diagnosing the disease and b) determine those who have one lesion (radiotherapy) vs more (systemic). Therefore knowing if a CT pelvis/spine or even conventional X-Pelvis/Spine will be sufficient to answer these questions, will be of much bigger interest. If the authors can conclude that conventional long bones/skull can be omitted, this will have a much better clinical relevance.

---

## Round 0.2 · Major Revisions

· Academic Editor

Major Revisions

Thanks for the revision. The manuscript was improved and is in reach of publication. Reviewer 3 raises an important issue that requires attention. Also, I still have a few issues:

1. I fail to find evidence for the second conclusion that larger mixed lesions are as pathognomonic as osteosclerotic lesions. Either remove this part of the conclusion or substantiate it.

2. The abstract statement " (97.4% vs. 86.8%, 994 vs. 276 lesions)" is odd. The number 276 is much smaller than 994, yet the SS still achieves 86.8% which is only a little reduction compared to 97.4%.

3. The correlation between serum levels and imaging features results in 42 Pearson tests. Repeated testing introduces the risk of having chance findings. Use appropriate statistical techniques (e.g. Bonferroni correction).

4. In the study population section a line says: '' were inλin 35 patients (92.1%; 35/38)'' . What do you mean by ''inλin''.

·

Basic reporting

No Comments

Experimental design

No Comments

Validity of the findings

No Comments

Additional comments

This manuscript has been well revised.

·

Basic reporting

No Comments

Experimental design

No Comments

Validity of the findings

No Comments

Additional comments

The author wrote “The conventional SS included skull, cervical/thoracic/lumbar spine, pelvis, bilateral humeri and femora. The CT scan range was from the upper margin of sternum to the ischium, and the slice thickness was 7mm.” in line 91, but stated “There was no patient whose bone lesion was detected by X-ray but not by CT.” in line 131. In the rebuttal letter to the first reviewer’s question “Did the authors find patients with bone lesion which was detected by X-ray but not by CT?”, the author wrote “Unfortunately, for the anatomical features (thoracic and lumbar spine and pelvis) covered by both X-ray and CT, there was no patient whose bone lesion was detected by X-ray but not by CT.” I think that the reviewer mean to ask whether there are some lesions which were detected by SS but not by CT. The answer should be “yes” from the article. The lesions in the skull and long bones which are out of field of CT can be only detected by SS, because the author said “The CT scan range was from the upper margin of sternum to the ischium”. The question cares about the lesions in the unique sites, but not the patients. Then, that is the unique usefulness of X-rays, compared to CT. The author should make some changes in those sentences.

·

Basic reporting

None

Experimental design

None

Validity of the findings

Review:
In general the response to the reviewer’s comments has been satisfactory. However some minor and major concerns remain:
Major:
On second re-evaluation of the correlation between skeletal markers and biochemical parameters, important considerations should be made. The authors correlated the number and size of lesions (on CT) with biomarkers. However these are systemic blood markers and are a result of lesions in the whole skeleton and not only due to axial skeleton. Therefore only “part” of the skeleton (CT) is correlated to blood markers. Therefore one might expect beforehand, that a good correlation will not be found. The authors should clearly state that thoracic-abdominal-pelvic skeletal disease burden does not correlate with biomarkers and should add this as a drawback. Furthermore, patients always have a mix between small < 5 mm, intermediate 5-10 and large > 10 lesions. Correlating the individual sizes to systemic biomarkers does not seem logical at all because one might expect beforehand that due to the heterogeneity of sizes a correlation is not to be expected. Also the number of lesions are not necessarily expected to correlate with biomarkers. This would be the case if for example if a patient has 20 small lesions compared to a patient with 5 large lesions. The volume of the lesions (therefore disease burden) would be expected to be a better correlator. Therefore a more scientific sound way of correlating with biomarkers, would be to determine the disease load (volume) of lesions and correlate this to biomarkers. This however is not practical at all. A method to do this more easily is to follow a similar approach like the RECIST criteria, where the sum of longest diameters are used. Therefore the sum of longest diameters of all lesions would give a much better idea of disease burden and would theoretically give a better correlation (if there is any). Important however again is to consider the fact that lesions in humeri, femura and skull are NOT included in the correlation. A similar importance is when comparing mixed lesions and sclerotic. Because patients never have pure mixed lesions and always a combination of mixed and sclerotic, it can be expected beforehand that a correlation will not be found. Again to compare the relative proportion of SUM (commulative length) of sclerotic vs mixed per patient might circumvent the inability to potentially show a correlation. Only in patients where the sum of mixed lesions is substantial, a sounder interpretation of the correlation albeit not with biomarkers can be made. It is advised that the authors look at this problem in depth.
Like in my prior comments, this study focuses on initial detection/diagnosis of POEMS not on management. Therefore it should clearly be mentioned that in these 38 patients, adding skull/appendix SS is of not use in the initial diagnosis. The recommendation can therefore be made that SS has NO role in these patients. Finding lesions in skull/appendix does not change diagnosis or initial management at all. The authors can included this in their conclusions. This is an important finding of their study. For patients with follow-up and assessment of treatment response, there might be a different scenario. The added benefit of SS in these patients should be evaluated in a future study.
Minor:
In the abstract, first sentence – the word “administering” should be changed to “management of”.
Results: Add percentages of the number of tumors identified in different locations, not only numbers.

---

## Round 0.3 · accepted · Accept

· Academic Editor

Accept

The authors adequately addressed the comments from the reviewers and me.

·

Basic reporting

Fine

Experimental design

Fine

Validity of the findings

Fine

Additional comments

The article is of sufficient quality for publications. The rebuttal is satisfactory.